# Germline molecular data in hereditary breast cancer in Brazil: Lessons from a large single-center analysis

**Renata Lazari Sandoval**[1][*], **Ana Carolina Rathsam Leite**[1], **Daniel Meirelles Barbalho**[1], **Daniele Xavier Assad**[1], **Romualdo Barroso**[1], **Natalia Polidorio**[1], **Carlos Henrique dos Anjos**[1], **Andréa Discaciati de Miranda**[2], **Ana Carolina Salles de Mendonça Ferreira**[3], **Gustavo dos Santos Fernandes**[1], **Maria Isabel Achatz**[4]

1 Department of Oncology, Hospital Sírio-Libanês, Brasília, Distrito Federal, Brazil, 2 Clínica da mama, Brasília, Distrito Federal, Brazil, 3 Oncoclínicas, Brasília, Distrito Federal, Brazil, 4 Department of Oncology, Hospital Sírio-Libanês, São Paulo, São Paulo, Brazil

☯ These authors contributed equally to this work.
* rsandoval.med@gmail.com

**Data Availability Statement:** All relevant data are within the manuscript and its Supporting

## Abstract

Brazil is the largest country in South America and the most genetically heterogeneous. The aim of the present study was to determine the prevalence of germline pathogenic variants (PVs) in Brazilian patients with breast cancer (BC) who underwent genetic counseling and genetic testing at a tertiary Oncology Center. We performed a retrospective analysis of the medical records of Brazilian patients with BC referred to genetic counseling and genetic testing between August 2017 and August 2019. A total of 224 unrelated patients were included in this study. Premenopausal women represented 68.7% of the cohort. The median age at BC diagnosis was 45 years. Multigene panel testing was performed in 219 patients, five patients performed single gene analysis or family variant testing. Forty-eight germline PVs distributed among 13 genes were detected in 20.5% of the patients (46/224). Eighty-five percent of the patients (91/224) fulfilled NCCN hereditary BC testing criteria. Among these patients, 23.5% harbored PVs (45/191). In the group of patients that did not meet NCCN criteria, PV detection rate was 3% (1/33). A total of 61% of the patients (28/46) harbored a PV in a high-penetrance BC gene: 19 (8.5%) *BRCA1/2*, 8 (3.5%) *TP53*, 1 (0.5%) *PALB2*. Moderate penetrance genes (*ATM*, *CHEK2*) represented 15.2% (7/46) of the positive results. PVs detection was statistically associated ($p<0.05$) with BC diagnosis before age 45, high-grade tumors, bilateral BC, history of multiple primary cancers, and family history of pancreatic cancer. According to the current hereditary cancer guidelines, 17.4% (39/224) of the patients had actionable variants. Nine percent of the patients (20/224) had actionable variants in non-*BRCA* genes, it represented 43.5% of the positive results and 51.2% of the actionable variants. Considering the observed prevalence of PVs in actionable genes beyond *BRCA1/2* (9%, 20/224), multigene panel testing may offer an effective first-tier diagnostic approach in this population.

Information files. The original data is available in an excel file.

**Funding:** The authors received no specific funding for this work.

**Competing interests:** The authors have declared that no competing interests exist.

## Introduction

Inherited germline pathogenic variants (PVs) in high or moderate penetrance breast cancer (BC) susceptibility genes are the underlying cause of approximately 15% of all BC cases [1, 2]. The implementation of preventive strategies may have an impact on cancer incidence and mortality in this high-risk population [3].

Several studies have demonstrated the cost-effectiveness of genetic testing, surveillance, prevention, and treatment strategies in PV carriers of cancer susceptibility genes [4]. Most studies are based on *BRCA1/2* carriers. Other genes included in the multigene panels used for the investigation of hereditary BC still lack sufficient data on penetrance, genotype-phenotype correlations, as well as benefits of intensive surveillance and risk reduction surgeries related to a mortality reduction [5, 6].

The prevalence of germline mutations varies widely, depending on the selected studied population. In unselected populations, the detection rate of clinically actionable pathogenic variants is low [7]. Studies in populations with known founder mutations may detect mutations in 1.1–4.5% of individuals not selected based on a personal or family history of cancer [8]. Genetic testing based on family history approaches has a moderate to high diagnostic accuracy in predicting the detection of an inherited mutation, depending on the selected risk prediction tool [9]. Nevertheless, 50% of *BRCA* mutation carriers are missed through family history criteria [10].

Even though there are published data on hereditary BC among Latin American countries [11, 12], few studies have included non-*BRCA* genes and multigene panel testing [13, 14]. Brazil is the largest country in South America and is the most genetically heterogeneous [15]. Most studies performed in Brazilian cohorts on hereditary breast cancer have been performed in the Southern and Southeastern parts of the country [13, 16]. For this reason, epidemiological data about the prevalence of cancer predisposition syndromes in other parts of the country are needed.

Brasília is the capital of Brazil, located in the central region of the country, and was founded in 1960. A large number of internal migrants came from all over the country to build it and ended up populating the city. Therefore, Brasília residents represent a unique sample of the Brazilian population. To the best of our knowledge, no previous study has described BC germline data using genetic testing in this region. The purpose of this study was to describe the clinical data and frequency of germline PVs in Brazilian BC patients from Brasília.

## Materials and methods

A total of 248 consecutive BC unrelated patients were referred for genetic counseling, between August 2017 and August 2019, at Hospital Sírio-Libanês (Brasília, Brazil), a tertiary oncology center. Ductal carcinoma in situ (DCIS) and invasive breast cancer were included in the study following NCCN criteria for further genetic evaluation. Twenty-four patients were excluded from the analysis because they did not undergo germline testing (Fig 1). A waiver of informed consent was approved by the Institutional Research Ethical Committee of the Hospital Sírio-Libanês (CAAE. 21735619.3.0000.5461). The Ethics Committee specifically reviewed and approved the protocol for our study.

Clinical information was retrospectively collected from the electronic medical records of patients. Electronic medical records were reviewed between November 2019 and January 2020. All personal and family history data were ascertained by institutional certified medical geneticists. All clinical and molecular data were de-identified before data sharing and analysis. The collected data included: age at cancer diagnosis; history of unilateral or bilateral breast cancer (synchronous or metachronous); histological subtype and tumor immunohistochemical profile; personal history of other primary cancers; family history of cancer (1st, 2nd, and 3rd

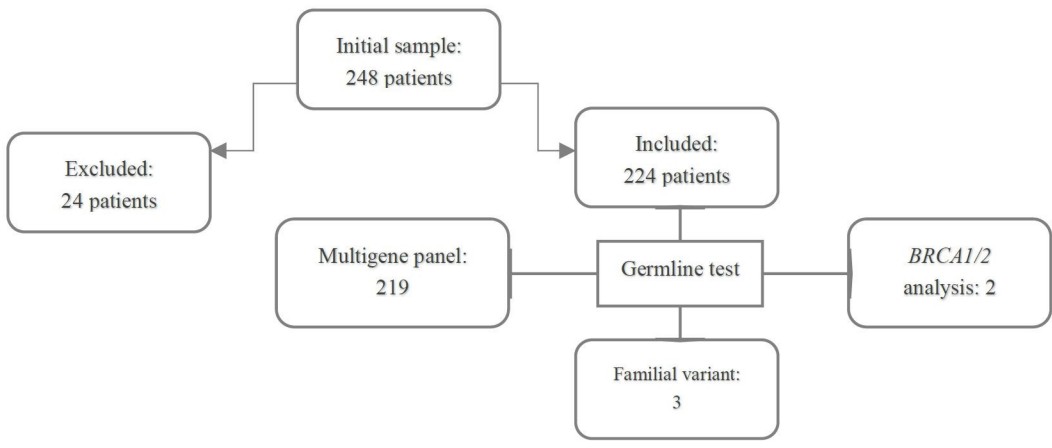

**Fig 1. Cohort selection.**

degree relatives); number of family members affected by BC; type of germline genetic test performed; number of analyzed genes, in case of multigene testing; and results of genetic testing, including PVs and variants of uncertain clinical significance (VUS). Detailed information about the gene panels studied in this cohort is described in the supplemental material.

All germline genetic tests were performed by commercial molecular diagnostic laboratories (S1 Table). Variants were classified according to the framework standardized by the American College of Medical Genetics and Genomics (ACMG) and Association for Molecular Pathology (AMP) [17].

## Statistical analyses

Continuous variables were tested for normality using the Kolmogorov-Smirnov and Shapiro-Wilk tests. Values are expressed as median and percentiles for non-parametric data, and as mean and standard deviation for parametric data. Categorical data are presented as absolute values and percentages and were tested using the Pearson $\chi2$ test and Fisher's exact test, when applicable.

Non-parametric data were compared using the Mann-Whitney U test for two independent samples or the Kruskal–Wallis test with a Müller-Dunn *post-hoc* test for three or more samples. Statistical significance was set at a $p \leq 0.05$. Statistical analyses were performed using SPSS 21.0 IBM®.

## Results

### Study population

A total of 224 patients with BC were included in this study. Baseline and demographic characteristics of the cohort are described in Table 1. Four patients were male (1.8%, 4/224). Most patients were diagnosed with primary BC under the age of 50 years (66.1%, 148/224), with a median age at diagnosis of 45 years (95% Confidence interval [CI], 38–53). Thirty-nine patients (17.4%, 39/224) had more than one primary cancer, of whom 79.4% (31/39) had two primary cancers and 20.5% (8/39) had 3 or more primary cancers. BC was the primary cancer diagnosed in 208 out of 224 patients (92.8%). The remaining sixteen patients had previously been diagnosed with cancer, including thyroid cancer (6/16), melanoma (4/16), central

**Table 1. Baseline and demographic characteristics.**

| Characteristics Total cohort = 224 | Patients with no germline pathogenic variants, n = 178 (%) | Patients with germline pathogenic variants, n = 46 (%) | P = value |
|---|---|---|---|
| **Gender** | | | |
| **Male** | 1 (0.6) | 3 (6.5) | **.007**** |
| **Female** | 177 (99.4) | 43 (93.5) | |
| **Number of primary cancers** | | | |
| **1** | 151 (84.8) | 34 (73.9) | **.046**** |
| **2** | 22 (12.4) | 9 (19.6) | |
| **≥ 3** | 5 (2.8) | 3 (6.5) | |
| **Age at breast cancer diagnosis** | | | |
| **≤ 31 years** | 11 (6.2) | 7 (15.2) | **.009**** |
| **32–44 years** | 68 (38.2) | 21 (45.7) | |
| **45–49 years** | 36 (20.2) | 1 (2.2) | |
| **≥ 50 years** | 63 (35.4) | 17 (37) | |
| **Menopausal status** | | | |
| **Premenopausal** | 113 (67.3) | 32 (74.4) | .366 |
| **Postmenopausal** | 55 (32.7) | 11 (25.6) | |
| **Missing data** | 10 | 3 | |
| **Laterality** | | | |
| **Unilateral** | 171 (96.1) | 40 (87) | **.018**** |
| **Bilateral** | 7 (3.9) | 6 (13) | |
| **Tumor histology** | | | |
| **NTS** | 118 (74.7) | 36 (87.8) | .345 |
| **DCIS** | 25 (15.8) | 4 (9.8) | |
| **ILC** | 13 (8.2) | 1 (2.4) | |
| **Others*** | 2 (1.1) | 0 (0) | |
| **Missing data** | 20 | 5 | |
| **Tumor Grade** | | | |
| **G3** | 48 (36.6) | 20 (64.5) | **.002**** |
| **G2** | 76 (58.0) | 7 (22.6) | |
| **G1** | 7 (5.3) | 4 (12.9) | |
| **Missing data** | 47 | 15 | |
| **Immunohistochemical profile** | | | |
| **HR+ HER2-** | 81 (54.4%) | 17 (48.6) | .384 |
| **HR+ HER2+** | 29 (19.5) | 8 (22.9) | |
| **HR- HER2+** | 14 (9.4) | 1 (2.9) | |
| **Triple negative** | 25 (16.8) | 9 (25.7) | |
| **Missing data** | 29 | 11 | |
| **Family history of cancer** | | | |
| **Negative or unknown** | 20 (11.2) | 3 (6.5) | .585 |
| **Positive** | 158 (88.8) | 43 (93.5) | |
| **Breast** | 99 (55.6) | 30 (65.2) | .215 |
| **Ovarian** | 12 (6.7) | 5 (10.9) | .681 |
| **Pancreatic** | 17 (9.5) | 11 (23.9) | **.048**** |
| **Prostate** | 46 (25.8) | 16 (34.8) | .441 |

Abbreviations: NTS- invasive carcinoma of no special type; DCIS, ductal carcinoma *in situ*; ILC, invasive lobular carcinoma; IHC, immunohistochemistry; G, grade; HR, hormonal receptors; HER2, human epidermal growth factor receptor 2.

*Others: sarcomatous cancer and malignant phyllodes.

**Statistically significant.

nervous system (2/16), kidney (1/16), uterine cancer (1/16), lymphoma (1/16), and pancreatic cancer (1/16).

Premenopausal BC was present in 68.7% of patients (145/211). Thirteen patients (5.8%, 13/224) had bilateral cancer, including 4 synchronous and 9 metachronous cancers. Invasive carcinoma of no special type (NTS) was present in 77.4% (154/199) of the patients, followed by ductal carcinoma *in situ* (DCIS) [14.6% (29/199)], invasive lobular carcinoma (ILC) [7% (14/199)], sarcomatous cancer or malignant phyllodes [1% (2/199)]. High-grade tumors (grade 3) represented 42% (68/162) of the cases, grade 2 51.2%, and grade 1 6.8%. Sixty-two cases had no description of tumor grade in the records. According to hormone receptor (HR) and human epidermal growth factor receptor 2 (HER2) status, HR+/HER2- tumors represented 53.3% of the cases (98/184), HR+/HER2+ 20.1% (37/184), HER2-enriched (HR-/HER2+) 8.1% (15/184), and 18.5% of cases were triple negative (HR-/HER2-). The immunohistochemical profile of forty cases was not available in the medical records.

Considering the family history of cancer, 89.7% (201/224) of the patients had, at least, one first- or second-degree family member affected by cancer. Sixty-four percent (129/201) of patients referred having family members with BC. Ovarian, pancreatic, and prostate cancer were present in 8.5% (17/201), 13.9% (28/201), and 30.9% (62/201) of family relatives, respectively. Seventy-one patients (31.7%) reported to have two or more first- or second-degree relatives with BC. Twenty-three patients (10.2%) did not provide any information about family history or had no known relatives with cancer.

## Frequency and spectrum of pathogenic germline variants

A total of 219 patients (97.7%) underwent multigene panel testing, 3 patients (1.3%) had family variant testing, and 2 patients (0.8%) underwent only *BRCA1/2* sequencing. Ten patients underwent multigene panel testing, including up to 50 genes, and 206 patients underwent gene testing with panels having 80 or more genes (S1 Table). The number of genes evaluated in the panel was not available for three patients. The number of tested genes varied according to the health insurance approval and the options of patients after pretest genetic counseling.

Forty-eight PVs were detected in 46 patients (20.5%, 46/224) (Fig 2). More details of the PVs detected are described in Table 2 and S2 Table. Two patients had more than one PV

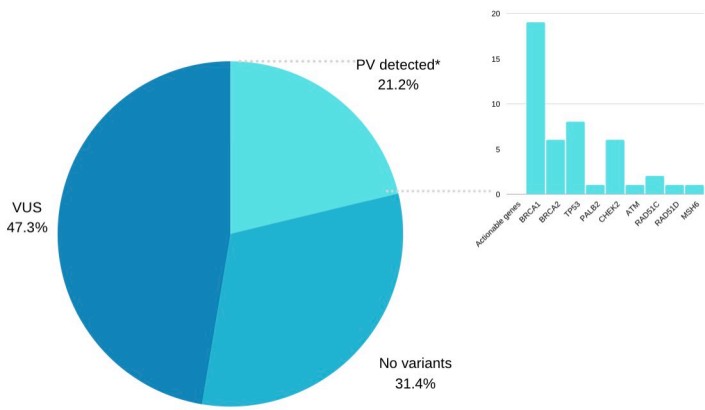

**Fig 2. Genetic test results.** *Included all PV detected (48/224). Two patients had more than one PV (*BRCA2*+monoallelic *MUTYH*; *BRCA1*+monoallelic *CTC1*). Actionable genes included *RAD51C* and *RAD51D* due to increased risk for ovarian cancer, as well as, *MSH6* for ovarian/endometrial/colorectal cancer. Monoallelic *MUTYH* PV were not included as actionable, although it is recommended colonoscopy at 40 years if there is a family history of colorectal cancer.

**Table 2. Pathogenic and likely pathogenic variants detected.**

| Gene | Variant | Classification | dbSNP or Variation ID | Number of patients |
|---|---|---|---|---|
| *ATM* | c.7913G>A (p.Trp2638*) | PV | rs377349459 | 1 |
| *BARD1* | c.176_177del (p.Glu59Alafs*8) | PV | rs1057517589 | 2 |
| *BRCA1* | del exons 8–19 | PV | Variation ID: 126018 | 1 |
| *BRCA1* | c.132C>G (p.Cys44Trp) | LP | rs876658362 | 1 |
| *BRCA1* | c.441+2T>A (splice donor) | PV | rs397509173 | 1 |
| *BRCA1* | c.791_794del (p.Ser264Metfs*33) | PV | rs80357707 | 1 |
| *BRCA1* | c.850C>T (p.Gln284Ter) | PV | rs397509330 | 1 |
| *BRCA1* | c.1115G>A (p.Trp372*) | PV | rs397508838 | 1 |
| *BRCA1* | c.1687C>T (p.Gln563*) | PV | rs80356898 | 2 |
| *BRCA1* | c.3598C>T (p.Gln1200*) | PV | rs62625307 | 1 |
| *BRCA1* | c.5177_5180delGAAA (p.Arg1726Lysfs*3) | | rs80357867 | 1 |
| *BRCA1* | c.5266dupC (p.Gln1756Profs*74) | PV | rs80357906 | 3 |
| *BRCA2* | c.156_157insAlu (p.Lys53Alafs) | PV | Variation ID: 126018 | 1 |
| *BRCA2* | c.1310_1313del (p.Lys437Ilefs*22) | PV | rs80359277 | 1 |
| *BRCA2* | c.3680_3681del (p.Leu1227Glnfs*5) | PV | rs80359395 | 1 |
| *BRCA2* | c.2512A>T (p.Lys838*) | PV | rs747578057 | 1 |
| *BRCA2* | c.5073dupA (p.Trp1692Metfs*3) | PV | rs80359479 | 1 |
| *BRCA2* | c.6405_6409del (p.Asn2135Lysfs*3) | PV | rs80359584 | 1 |
| *CHEK2* | c.319+2T>A (splice donor) | LP | rs587782401 | 1 |
| *CHEK2* | c.349A>G (p.Arg117Gly) | LP | rs28909982 | 2 |
| *CHEK2* | c.593-1G>T (splice acceptor) | LP | rs786203229 | 1 |
| *CHEK2* | c.846+1G>C (splice donor) | LP | rs864622149 | 1 |
| *CHEK2* | c.1008+2T>G (splice donor) | LP | rs1555915295 | 1 |
| *MUTYH* | c.305-1G>C (splice acceptor) | PV | rs372267274 | 1 |
| *MUTYH* | c.933+3A>C (Intronic) | PV | rs587780751 | 1 |
| *MUTYH* | c.1187G>A (p.Gly396Asp) | PV | rs36053993 | 2 |
| *MSH6* | c.1519dupA (p.Arg507Lysfs*8) | PV | rs876658881 | 1 |
| *PALB2* | Deletion exon 2–3 | PV | - | 1 |
| *RAD51C* | c.709C>T (p.Arg237*) | PV | rs770637624 | 2 |
| *RAD51D* | c.694C>T (p.Arg232*) | PV | rs587780104 | 1 |
| *RECQL4* | c.1166_1167del (p.Cys389Phefs*33) | PV | rs34134064 | 1 |
| *TP53* | Partial deletion exon 5 | PV | - | 1 |
| *TP53* | c.733G>A (p.Gly245Ser) | PV | rs28934575 | 1 |
| *TP53* | c.1010 G>A (p.Arg337His) | PV | rs121912664 | 6 |

Abbreviations: PV, Pathogenic variant; LP, Likely pathogenic.

detected: PV in *BRCA2* and a monoallelic PV in *MUTYH*, PV in *BRCA1* and a monoallelic PV in *CTC1*.

A total of 191 patients (85.3%, 191/224) fulfilled NCCN hereditary BC testing criteria. Among these patients, 23.5% harbored PVs (45/191). In the group of patients that did not meet NCCN criteria, PV detection rate was 3% (1/33) (S3 Table).

PVs were distributed among 13 genes. Sixty-one percent of patients (28/46) had PVs in a high-penetrance gene for BC: 19 (8.5%) in *BRCA1/2*, 8 (3.5%) in *TP53*, and 1 (0.5%) in *PALB2*. Moderate penetrance genes for BC (*ATM*, *CHEK2*) represented 15.2% (7/46) of the positive results: 6 (13%) patients had PV in *CHEK2* and 1 (2.1%) in *ATM*. PVs in genes considered to have a potential increased risk of BC (*BARD1*, *RAD51C*, *RAD51D*) or an unknown risk/

insufficient data (*MUTYH*, *MSH6*, *RECQL4*) were found in 23.9% (11/46) of patients. According to current guidelines [18], 39 of the 224 tested patients (17.4%) had actionable variants. Twenty out of 39 (51.2%) actionable variants were found in non-*BRCA* genes.

## Correlation between test positivity, germline genotype, and clinical data

The diagnosis of bilateral BC occurred in 13% (6/46) of patients who harbored a germline PV (2 *BRCA2* carriers, 2 *CHEK2*, 1 *BARD1*, and 1 *RAD51C*). Bilateral BC was predictive of a positive test (p = 0.018). Considering all the cases of bilateral BC (13/224), 31% (4/13) harbored PVs in non-*BRCA* genes. Two patients had ipsilateral breast tumors; however, it was not possible to determine whether it was a new primary tumor or a local recurrence. Both had a negative test result.

The majority of PVs were nonsense or frameshift (24/48, 50%). There were 14 frameshift variants (29.2%), 10 nonsense variants (20.8%), 13 missense variants (27%) and 7 splice site variants (14.6%). The remaining four variants (4/48, 8.4%) were pathogenic copy number variations (CNVs), including the diagnosis of one Alu insertion in *BRCA2* (c.156_157insAlu, a Portuguese founder mutation), one case of *BRCA1* exons 8–19 deletion, one *PALB2* exon 2–3 deletion, and one *TP53* partial deletion of exon 5 (possibly mosaic). The partial deletion in *TP53*, possibly in mosaic, was not confirmed using fibroblast genetic testing due to patient death.

A young age at BC diagnosis (< 45 years) was statistically associated with PV detection (p = 0.040). Most of the patients with a positive test result had a cancer diagnosis before the age of 50 years (63%, 29/46). It included 45.6% of patients aged 32–44 years (21/46), 15.2% (7/46) who were under 31 years old, and 2.2% (1/46) aged 45–49 years. High grade tumors were associated with a higher probability of PV detection (p = 0.002). Although premenopausal BC alone was not statistically associated with PV detection, 62% of premenopausal women with high-grade BC had a PV identified in the genetic test (p = 0.008).

Multiple primary tumors were more common among patients with a positive test compared to patients with a negative/VUS test result (26.1% vs. 15.2%, p = 0.042). Multiple family members affected by breast, prostate, or pancreatic cancer were associated with a higher probability of PV detection (p = 0.010). There was a statistical association between PVs and a family history of ovarian cancer only for PVs in the *BRCA1* gene (p < 0.001).

At least one VUS was described in the genetic test reports of 140 patients (62.5%) (S4 Table). Most patients had only 1 VUS detected (82/140; 58,6%), but in approximately 41,4% (58/140) of cases, 2–6 VUS were described. Excluding patients with VUS and confirmed PV (33/140), VUSs were described in 107 patients. The frequency of VUS increased according to the number of genes included in the genetic test (p = 0.006).

## Discussion

Hereditary BC has significant genetic heterogeneity. The National Comprehensive Cancer Network (NCCN) guidelines (version 1.2021) recommend genetic evaluation, including the genes *BRCA1*, *BRCA2*, *TP53*, *ATM*, *CDH1*, *CHEK2*, *NBN*, *NF1*, and *PALB2* for high-risk breast and/or ovarian cancer patients [18]. There is cumulative evidence that variants in *BARD1*, *BRIP1*, *MSH2*, *MLH1*, *MSH6*, *PMS2*, *RAD51C*, and *RAD51D* may also be implicated in hereditary BC [19–22]. Next-generation sequencing (NGS) technologies have been an effective method in a multigene testing scenario [23]. In addition, the costs of DNA sequencing have been decreasing significantly with the use of NGS [24]. Notwithstanding, there are global disparities in genetic testing accessibility.

Genetic testing is not accessible for the majority of the population from developing and underdeveloped countries. Although there is an international current debate about BC genetic testing based on clinical criteria versus BC universal testing [25–27], Brazil and other Latin America countries face limited access. The main reported barriers are related to the lack of structured genetic counseling and genetic testing networks, insufficient number of trained professionals in high risk cancer assessment, absence of genetic testing access in the public health system, limited health insurance coverage, costs of genetic testing and lack of national policies [28, 29].

Recent data, collected in 2020 by the Brazilian National Agency of Supplementary Health, estimated that only 24% of the Brazilian population has access to health insurance [30]. Therefore, more than 163 million people depend on the Brazilian national public health system, which has no access to genetic tests for the investigation of cancer predisposition syndromes. Since 2018, supplementary health care must cover germline genetic testing, according to some clinical criteria (much more restricted than those described in NCCN guidelines). The present study included patients from supplementary health care and, for this reason, 90% of the initial sample had access to genetic testing. A total of 17.4% of the tested patients (39/224) had actionable variants according to the current NCCN guidelines [18].

The detection rate of actionable PVs varies widely depending on the selected studied population and the genetic testing approach (founder mutations, single gene analysis, copy number variation evaluation, multigene panel testing). The present cohort consisted of consecutive BC patients referred to genetic counseling selected due to the suspicion of hereditary BC. The overall detection rate of PVs was 20.5% (46/224). According to NCCN criteria, 85.3% (191/224) of our cohort met hereditary BC testing criteria, among these patients 23.6% (45/191) harbored PVs. Whereas, among 33 patients that did not met NCCN criteria, only one harbored a PV (3%). The detection rate of PVs was similar to other multigene panel testing studies based on hereditary breast and ovarian cancer (HBOC) criteria [5, 13, 14, 31–33].

Fifty-seven percent of the patients with a positive test result harbored PVs in high-penetrance BC genes (41.3% in *BRCA1/2*, 17.4% in *TP53*, and 2.2% in *PALB2)*, 15.2% in moderate penetrance BC genes (2.2% in *ATM* and 13% in *CHEK2*), and 23.9% in genes considered to be associated with a BC potential increased risk or an unknown risk. Nine percent of the patients (20/224) had actionable variants in non-*BRCA* genes, representing 43.5% of the positive results and 51.2% of the actionable variants.

The detection rate of PVs in moderate penetrance BC genes varies from 2 to 8% [5, 33–35]. Our cohort was enriched by PVs in moderate penetrance BC genes (15.2% of positive results, 9% of overall genetic tests). Another Brazilian study with individuals from the Northeast of the country also found a high prevalence of PVs (32% of positive results) in moderate penetrance BC genes (12). These findings suggest that Brazilian patients should have access to multigene panel testing.

Although there is a higher frequency of *CHEK2* founder mutations (c.1100delC, c.470T>C) in European ancestry populations, these mutations were not observed in our cohort. Other Brazilian studies have also showed no enrichment of *CHEK2* European founder mutations in BC Brazilian patients [36–39]. The majority of *CHEK2* PVs reported in our study affected RNA splicing (Table 2).

There are limited studies from Brazil and Latin American countries with BC germline characterization by multigene panel testing. A Brazilian BC study from the Northeast of the country showed a PV frequency of 17% (27/157), 68% (13/19) harbored a PV in *BRCA1/2* genes and 32% (6/19) in moderate penetrance BC genes (12). Most of these patients were tested using a 33-gene panel. A research group from the Brazilian Southeast region performed a 21-gene panel in 95 women with a personal history of BC or HBOC clinical suspicion based

on family history criteria [14]. Twenty-three percent of the patients harbored a PV in *BRCA1/2* and *TP53* genes. Eighty-five women from Colombia, meeting HBOC criteria, had germline testing with a commercial 25-gene hereditary cancer panel [33]. Twenty-two percent of the patients (19/85) harbored a PV in a cancer susceptibility gene.

NGS approaches may have limited ability to detect copy number variations (CNVs). Supplemental methods, along with NGS analysis, are required to validate CNV detection from NGS panels [40]. In the present study, 8.7% of the patients with PV (4/46) had pathogenic CNVs detected using multigene panel testing: one Alu insertion in *BRCA2* (c.156_157insAlu), one *BRCA1* exon 8–19 deletion, one *PALB2* exon 2–3 deletion, and one *TP53* partial deletion of exon 5. Previously, Ewald *et al.* (2016) observed a 3.4% prevalence of *BRCA1/2* CNVs among 145 unrelated Brazilian individuals at risk of HBOC syndrome, which included three cases of Alu insertion in *BRCA2* (c.156_157insAlu) [41].

The prevalence of Li-Fraumeni syndrome (LFS) in our cohort was 3.5% (8/224). Giacomazzi *et al.* (2014) reported a 3.4% prevalence of p.R337H in Brazilian women diagnosed with BC who met the criteria for HBOC [42]. Considering all the patients with a detected PV in the present cohort, LFS represented 17.4% (8/46) of the PV carriers. Six out of eight patients diagnosed with LFS in this study carried the Brazilian *TP53* p.R337H variant, which corresponds to a prevalence of 2.7% (6/224). Similarly, Hahn *et al.* (2018) observed a frequency of the p.R337H variant in 2.5% (6/239) of Brazilian patients with BC diagnosed before age 46, unselected by family history [43]. It is well known that LFS has a higher prevalence in Brazil due to the *TP53* founder mutation c.1010G>A (p.Arg337His), also known as p.R337H [44]. These findings raise the question of whether this prevalence justifies routine screening of all Brazilian women with BC, despite the Chompret criteria. In the Ashkenazi Jewish population, in which *BRCA1/2* founder mutations are present in 2.5% of the individuals, cost-effectiveness studies have implied that population testing is justified [45].

Patients with cancer predisposition syndromes have a higher risk of developing a second primary BC, especially for *BRCA 1/2* mutation carriers [46]. Nevertheless, 8–36% of patients with bilateral BC harbor PVs in other genes beyond *BRCA* [32, 47, 48]. Bilateral BC represented 5.8% (13/224) of our cohort. Thirty-one percent (4/13) of patients with bilateral BC harbored PVs in non-*BRCA* genes (*CHEK2*, *BARD1* and *RAD51C*). These results are in concordance with previous studies [32, 47].

The concepts of clinical validity and clinical utility are important for the implementation of multi-gene testing and the development of guidelines for hereditary BC [6]. Genes classified as BC potential increased risk or unknown risk are not expected to change BC screening or management; nevertheless, they may provide information for high-risk assessment or risk reduction surgeries for other cancer sites. Four out of 46 carriers of PVs (8.7%) harbored a PV in *RAD51C*, *RAD51D*, and *MSH6*. *RAD51C* and *RAD51D* confer higher risks of ovarian cancer and *MSH6*, for endometrial, ovarian, and colorectal cancer. Monoallelic *MUTYH* PVs were not included as actionable in the current study, although colonoscopy is recommended at 40 years, if there is a family history of colorectal cancer. Despite the fact that monoallelic *MUTYH* PV is not an uncommon finding in BC patients undergoing multigene testing, it is not associated with BC risk [49, 50].

This study had several limitations. It was retrospective and had a limited sample size compared to multigene testing studies around the world. However, it represents the largest Brazilian BC cohort from a single institution tested using a multigene panel for hereditary BC [13, 14]. It is also the first BC germline characterization from the Center-West of the country. These findings require validation in other cohorts. The study was conducted at a private cancer center and, for this reason, the assessed population may differ from that of the general community. Furthermore, it consisted of a high-risk population for hereditary cancer with a median

age of 45 years at BC diagnosis, mostly composed of patients with premenopausal BC (68.7%), a positive family history for cancer (89.7%), and a personal history of multiple primary cancers (17.4%).

## Conclusion

This is the first study with germline molecular data from patients affected by BC in the Center-West of Brazil. We found a 20.5% prevalence of PVs. Seventeen percent of these patients had actionable variants according to the current guidelines. Nine percent of the patients had actionable variants in non-*BRCA* genes, representing 43.5% of the positive results and 51.2% of the actionable variants. Eighty-five of the patients fulfilled NCCN hereditary BC testing criteria, among these patients 23.5% harbored PVs. In our study, BC prior to 45 years, multiple primary cancers, high-grade tumors, bilateral BC, and a family history of pancreatic cancer were features associated with a higher probability of PV detection. Multigene panel testing may offer an effective first-tier diagnostic approach in this high-risk population.

## Supporting information

**S1 Table. Characteristics of genetic tests performed.**
(DOCX)

**S2 Table. Clinical characteristics of patients with pathogenic and likely pathogenic variants.**
(DOCX)

**S3 Table. Number of patients who meet current genetic testing criteria according to the genetic test result.**
(DOCX)

**S4 Table. Variants of uncertain significance detected.**
(DOCX)

## Author Contributions

**Conceptualization:** Renata Lazari Sandoval.

**Data curation:** Renata Lazari Sandoval, Ana Carolina Rathsam Leite, Daniel Meirelles Barbalho, Daniele Xavier Assad, Romualdo Barroso, Natalia Polidorio, Carlos Henrique dos Anjos, Andréa Discaciati de Miranda, Ana Carolina Salles de Mendonça Ferreira.

**Formal analysis:** Renata Lazari Sandoval, Ana Carolina Rathsam Leite.

**Investigation:** Renata Lazari Sandoval, Ana Carolina Rathsam Leite.

**Methodology:** Renata Lazari Sandoval, Ana Carolina Rathsam Leite.

**Project administration:** Renata Lazari Sandoval, Ana Carolina Rathsam Leite.

**Supervision:** Renata Lazari Sandoval, Romualdo Barroso, Gustavo dos Santos Fernandes, Maria Isabel Achatz.

**Visualization:** Ana Carolina Rathsam Leite, Gustavo dos Santos Fernandes.

**Writing – original draft:** Renata Lazari Sandoval, Ana Carolina Rathsam Leite.

**Writing – review & editing:** Renata Lazari Sandoval, Ana Carolina Rathsam Leite, Daniel Meirelles Barbalho, Daniele Xavier Assad, Romualdo Barroso, Natalia Polidorio, Carlos

Henrique dos Anjos, Andréa Discaciati de Miranda, Ana Carolina Salles de Mendonça Ferreira, Gustavo dos Santos Fernandes, Maria Isabel Achatz.

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
