## [Decision Letter · Decision Letter 0]

9 Nov 2020

PONE-D-20-32343

Germline molecular data in hereditary breast cancer in Brazil: lessons from a large single-center analysis.

PLOS ONE

Dear Dr. Sandoval,

Thank you for submitting your manuscript to PLOS ONE. After careful consideration, we feel that it has merit but does not fully meet PLOS ONE’s publication criteria as it currently stands. Therefore, we invite you to submit a revised version of the manuscript that addresses the points raised during the review process.

1.  It is unusual that missense variants make up the majority of pathogenic variants (unless we are considering TP53).  More details should be included on how variants were classified.  Furthermore, a table of all pathogenic and VUS found in the paper should be included in the supplemental materials.

2.  It is not clear why DCIS is considered as a breast cancer diagnosis.  More rationale for this should be included or consider only using primary invasive breast cancer as a diagnosis.

3.  On Figure 2, the "no PV or VUS" group should be renamed as Likely benign or no variants.  Figure 2 might be improved by using color.

4. In the Discussion expand discussion of the other Brazilian studies that have been published and how this study compares.  Also include any additional studies of other nearby countries.

5.  Deposit sequence variants found into a publicly accessible database.

We look forward to receiving your revised manuscript.

Kind regards,

Amanda Ewart Toland, Ph.D.

Academic Editor

PLOS ONE

Journal Requirements:

2. Thank you for including your ethics statement: "This research involved human participants. A waiver of informed consent was approved by the Institutional Research Ethical Committee (CAAE. 21735619.3.0000.5461)."   

3. In the ethics statement in the manuscript and in the online submission form, please provide additional information about the patient records/samples used in your retrospective study, including: a) whether all data were fully anonymized before you accessed them; b) the date range (month and year) during which patients' medical records/samples were accessed.

4.  Please ensure that you include a title page within your main document. We do appreciate that you have a title page document uploaded as a separate file, however, as per our author guidelines (http://journals.plos.org/plosone/s/submission-guidelines#loc-title-page) we do require this to be part of the manuscript file itself and not uploaded separately.

5. Please remove your figures from within your manuscript file, leaving only the individual TIFF/EPS image files, uploaded separately.  These will be automatically included in the reviewers’ PDF.

Reviewers' comments:

Reviewer's Responses to Questions

**Comments to the Author**

1. Is the manuscript technically sound, and do the data support the conclusions?

Reviewer #1: Yes

Reviewer #2: Partly

2. Has the statistical analysis been performed appropriately and rigorously? 

Reviewer #1: Yes

Reviewer #2: Yes

3. Have the authors made all data underlying the findings in their manuscript fully available?

Reviewer #1: No

Reviewer #2: Yes

4. Is the manuscript presented in an intelligible fashion and written in standard English?

Reviewer #1: Yes

Reviewer #2: Yes

5. Review Comments to the Author

Reviewer #1: October 25th, 2020

Comments to authors:

General:

The MS “Germline molecular data in hereditary breast cancer in Brazil: lessons from a large single-center analysis” by Renata Lazari Sandoval, M.D., M.Sc et al, describes the results using a panel of genes analyzed by NGS in breast cancers patients, concluding the benefits for use of panel genes. This MS has interesting regional results, although, as disclosed by the authors, the number of cases listed, nowadays, are a clear limitation if a strong conclusion should be draft.

A few comments to be considered:

a) Supplementary material table, is difficult to follow since it is based on the patient ID rather than in a field related to the topic of the MS (i.e. pathogenic variant, or IHC o whatever is chosen)

b) In Figure 2. a straight category might be more adequate (like “Benign”) rather that “No PV or VUS”

c) There are representative studies in Brazil that were ignored in this MS, and the utility is obvious to contrasting the results of the cohorts. Although the authors disclosed:

This study had several limitations. It was retrospective and had a limited sample size. However, it represents the largest Brazilian BC cohort from a single institution tested using a multigene panel for hereditary BC (12,13). These findings require validation in other cohorts. The study was conducted at a private cancer center and, for this reason, the assessed population may differ from that of the general community. Furthermore, it consisted of a high-risk population for hereditary cancer with a median age of 45 years at BC diagnosis, mostly composed of patients with premenopausal BC (68.7%), a positive family history for cancer (89.7%), and a personal history of multiple primary cancers (17.4%).

It might be a good improvement to discuss comparing with larger cohorts, even of other regions, from the large country Brazil.

d) Discussing the application of the NCCN guidelines upon the findings in the present work, may be interesting

e) A good idea is to deposit the variants in a public database for the free access to the worldwide scientific community

Reviewer #2: Methods: I would recommend using primary breast cancer as an inclusion criterion.

Results:

Study population: I would recommend excluding the remaining sixteen patients who had previously been diagnosed with other cancer because according to the title this paper is focused on breast cancer.

Table 1: I would recommend including an extra column with the information of the total cohort for each item.

Frequency and Spectrum of Pathogenic Germline Variants: I recommend including in supplementary data analyzed genes in each group of patients. I would also recommend presenting in the article text a table with the PV detected.

Correlation between test positivity, germline genotype, and clinical data: I recommend including a table with the VUS detected in supplementary data.

6. PLOS authors have the option to publish the peer review history of their article (what does this mean?). If published, this will include your full peer review and any attached files.

Reviewer #1: No

Reviewer #2: **Yes: **Laura Cifuentes-C

---

## [Author Response · Author response to Decision Letter 0]

25 Dec 2020

Manuscript PONE-D-20-32343

Response to reviewers

Dear Dr. Toland, 

Thank you for giving us the opportunity to submit a revised version of the manuscript “Germline molecular data in hereditary cancer in Brazil: lessons from a large single-center analysis”. We are grateful for the time and effort that you and the reviewers dedicated to providing feedback on our manuscript. All the comments provided the possibility of relevant improvements to our manuscript.

We have incorporated most of the suggestions made by the reviewers and clarified all the points raised during the review process. Authors´ responses are described in blue in a point-by-point response to the reviewers’ comments and concerns. In the first section the editor´s points were addressed, in the second section Reviewer 1 and in the third section Reviewer 2.

Changes performed in the revised manuscript are highlighted in yellow. Page numbers refer to the revised manuscript file with tracked changes.

Section 1- Editor´s comments to the authors: 

1. It is unusual that missense variants make up the majority of pathogenic variants (unless we are considering TP53).

Author response: We revised all the 48 likely pathogenic/pathogenic variants observed in this study: 10 nonsense, 14 frameshift, 4 copy number variants, 13 missense, 7 splice site. We thank you for the opportunity to correct this error, missense variants did not make up the majority of pathogenic variants, it represented 27% of pathogenic variants (13/48). Nonsense and frameshift variants, together, represented the majority of pathogenic variants (24/48, 50%). 

Changes in the text of the revised manuscript: page 11 line 160, “The majority of PVs were nonsense or frameshift (24/48, 50%). There were 14 frameshift variants (29.2%), 10 nonsense variants (20.8%), 13 missense variants (27%) and 7 splice site variants (14.6%). The remaining four variants (4/48, 8.4%) were pathogenic copy number variations (CNVs), including the diagnosis of one Alu insertion in BRCA2 (c.156_157insAlu, a Portuguese founder mutation), one case of BRCA1 exons 8-19 deletion, one PALB2 exon 2-3 deletion, and one TP53 partial deletion of exon 5 (possibly mosaic). The partial deletion in TP53, possibly in mosaic, was not confirmed using fibroblast genetic testing due to patient death.”

More details should be included on how variants were classified. Furthermore, a table of all pathogenic and VUS found in the paper should be included in the supplemental materials.

Author response: All germline genetic tests were performed by commercial molecular diagnostic laboratories. We created the table S1 (supplementary material) with the characteristics of the genetic tests performed. Variant classification performed by each commercial laboratory was specified. 

Variants were classified according to the framework standardized by the American College of Medical Genetics and Genomics (ACMG) and Association for Molecular Pathology (AMP). The majority of the genetic tests (196/219, 90%) were performed in the laboratory Invitae (San Francisco, California, USA). The Invitae Clinical Genomic Group also use the Sherloc framework for variant classification besides the ACMG-AMP variant classification guidelines.

All likely pathogenic/pathogenic were described in Table 2 in the manuscript. More detailed information about each patient that harbored a PV was described in S2 (supplementary material). All VUS were described in S4 (supplementary material). A column in Table 2 and S4 describes the variant identification as rsID (The Single Nucleotide Polymorphism Database identification, a free public archive for genetic variation) or Clinvar variant identifier number (variation ID). 

Changes in the the text of the revised manuscript: page 4 line 74, “All germline genetic tests were performed by commercial molecular diagnostic laboratories (S1 Table). Variants were classified according to the framework standardized by the American College of Medical Genetics and Genomics (ACMG) and Association for Molecular Pathology (AMP) (16).”

2. It is not clear why DCIS is considered as a breast cancer diagnosis. More rationale for this should be included or consider only using primary invasive breast cancer as a diagnosis. 

Author response: Although DCIS is a non-invasive or pre-invasive breast cancer, the National Comprehensive Cancer Network guidelines for Genetic/Familial High-Risk Assessment (https://www.nccn.org/professionals/physician_gls/default.aspx) include both invasive and ductal carcinoma in situ breast cancers as a criteria for further genetic evaluation (footnote d, version 1.2021, page 17 from NCCN guideline). Some studies also have showed that DCIS should be a criteria for hereditary breast cancer risk assessment as it seems to occur equally in sporadic and hereditary breast cancer cases. In mutation carriers it seems to occur at an earlier age. 

References:

Hwang ES, McLennan JL, Moore DH, Crawford BB, Esserman LJ, Ziegler JL. Ductal carcinoma in situ in BRCA mutation carriers. J Clin Oncol. 2007 Feb 20;25(6):642-7. doi: 10.1200/JCO.2005.04.0345.

Bayraktar S, Elsayegh N, Gutierrez Barrera AM, et al. Predictive factors for BRCA1/BRCA2 mutations in women with ductal carcinoma in situ. Cancer. 2012 Mar 15;118(6):1515-22. doi: 10.1002/cncr.26428.

Liu Y, Ide Y, Inuzuka M, et al. BRCA1/BRCA2 mutations in Japanese women with ductal carcinoma in situ. Mol Genet Genomic Med. 2019 Mar;7(3):e493. doi: 10.1002/mgg3.493.

Changes in the text of the revised manuscript: page 4 line 56, “Ductal carcinoma in situ (DCIS) and invasive breast cancer were included in the study following NCCN criteria for further genetic evaluation.” 

3. On Figure 2, “no PV or VUS” group should be renamed as Likely benign or no variants. Figure 2 might be improved by using color.

Author response: “no PV or VUS” group was renamed as “no variants”. Figure 2 was improved by using color.

4. In the Discussion expand discussion of the other Brazilian studies that have been published and how this study compares. Also include any additional studies of other nearby countries.

Author response: Most germline breast cancer studies in Latin America, including Brazil, focused on BRCA genes and used different testing approaches (founder mutation analysis, single gene analysis, copy number variation evaluation, multigene panel testing). For this reason, it was difficult to compare results from Brazil and nearby countries. The best scenario would be the comparison of our data to multigene panel testing studies. Nevertheless, after your suggestion we performed a new revision of the literature with the proposed view. Pubmed search pointed out only 3 studies with multigene panel testing (2 from Brazil and 1 from Colombia). (in the document "Response to reviewers" attached, there is a table with the information about these publications)

The reference 11 in our manuscript [Urbina-Jara LK, et al. Landscape of Germline Mutations in DNA Repair Genes for Breast Cancer in Latin America: Opportunities for PARP-Like Inhibitors and Immunotherapy. Genes (Basel). 2019 Oct 10;10(10):786], shows an excellent review of all the publications from Latin America countries related to BRCA and non-BRCA genes testing. Brazil is the country with most of the published data, in contrast with the majority of the other countries, many Latin America countries do not have any published data at all.

Here we illustrate the heterogeneity of genetic testing approaches in the most relevant Brazilian published studies:

Non-Multigene panel testing Brazilian BC data is illustrated in a table in the document attached "Response to reviewers".

We rewrote the discussion trying to incorporate suggestions.

Changes in the text of the revised manuscript: page 12 line 193-200, “Notwithstanding, there are global disparities in genetic testing accessibility. 

Genetic testing is not accessible for the majority of the population from developing and underdeveloped countries. Although there is an international current debate about BC genetic testing based on clinical criteria versus BC universal testing (24–26), Brazil and other Latin America countries face limited access. The main reported barriers are related to the lack of structured genetic counseling and genetic testing networks, insufficient number of trained professionals in high risk cancer assessment, absence of genetic testing access in the public health system, limited health insurance coverage, costs of genetic testing and lack of national policies (27,28)”

Changes in the text of the revised manuscript: page 13-14 line 209-240, “The detection rate of actionable PVs varies widely depending on the selected studied population and the genetic testing approach (founder mutations, single gene analysis, copy number variation evaluation, multigene panel testing). The present cohort consisted of consecutive BC patients referred to genetic counseling selected due to the suspicion of hereditary BC. The overall detection rate of PVs was 20.5% (46/224). According to NCCN criteria, 85.3% (191/224) of our cohort met hereditary BC testing criteria, among these patients 23.6% (45/191) harbored PVs. Whereas, among 33 patients that did not met NCCN criteria, only one harbored a PV (3%). The detection rate of PVs was similar to other multigene panel testing studies based on hereditary breast and ovarian cancer (HBOC) criteria (5,12,13,30–32).

Fifty-seven percent of the patients with a positive test result harbored PVs in high-penetrance BC genes (41.3% in BRCA1/2, 17.4% in TP53, and 2.2% in PALB2), 15.2% in moderate penetrance BC genes (2.2% in ATM and 13% in CHEK2), and 23.9% in genes considered to be associated with a BC potential increased risk or an unknown risk. Nine percent of the patients (20/224) had actionable variants in non-BRCA genes, representing 43.5% of the positive results and 51.2% of the actionable variants. 

The detection rate of PVs in moderate penetrance BC genes varies from 2 to 8% (5,32–34). Our cohort was enriched by PVs in moderate penetrance BC genes (15.2% of positive results, 9% of overall genetic tests). Another Brazilian study with individuals from the Northeast of the country also found a high prevalence of PVs (32% of positive results) in moderate penetrance BC genes (12). These findings suggest that Brazilian patients should have access to multigene panel testing. 

Although there is a higher frequency of CHEK2 founder mutations (c.1100delC, c.470T>C) in European ancestry populations, these mutations were not observed in our cohort. Other Brazilian studies have also showed no enrichment of CHEK2 European founder mutations in BC Brazilian patients (35–38). The majority of CHEK2 PVs reported in our study affected RNA splicing (Table 2). 

There are limited studies from Brazil and Latin American countries with BC germline characterization by multigene panel testing. A Brazilian BC study from the Northeast of the country showed a PV frequency of 17% (27/157), 68% (13/19) harbored a PV in BRCA1/2 genes and 32% (6/19) in moderate penetrance BC genes (12). Most of these patients were tested using a 33-gene panel. A research group from the Brazilian Southeast region performed a 21-gene panel in 95 women with a personal history of BC or HBOC clinical suspicion based on family history criteria (13). Twenty-three percent of the patients harbored a PV in BRCA1/2 and TP53 genes. Eighty-five women from Colombia, meeting HBOC criteria, had germline testing with a commercial 25-gene hereditary cancer panel (32). Twenty-two percent of the patients (19/85) harbored a PV in a cancer susceptibility gene.” 

Changes in the text of the revised manuscript: page 15 line 275-283, ”This study had several limitations. It was retrospective and had a limited sample size compared to multigene testing studies around the world. However, it represents the largest Brazilian BC cohort from a single institution tested using a multigene panel for hereditary BC (12,13). It is also the first BC germline characterization from the Center-West of the country. These findings require validation in other cohorts. The study was conducted at a private cancer center and, for this reason, the assessed population may differ from that of the general community. Furthermore, it consisted of a high-risk population for hereditary cancer with a median age of 45 years at BC diagnosis, mostly composed of patients with premenopausal BC (68.7%), a positive family history for cancer (89.7%), and a personal history of multiple primary cancers (17.4%).” 

5. Deposit sequence variants found into a publicly accessible database.

Author response: All variants were included in publicly accessible databases such as ClinVar from the National Center for Biotechnology information, by the commercial laboratories. Invitae also created a free access database called ClinVitae (clinvitae.invitae.com), which aggregates data from ClinVar (including variants contributed by Invitae), Emory Genetics Variant Classification catalog, ARUP Mutation Databases, Carvaer Mutation Database, Kathleen Cunninham Foundation Consortium for Research in Familial Breast Cancer. We also included in Table 2 and S4 the rsID related to the Single Nucleotide Polymorphism Database identification or the Clinvar variant identifier number (variation ID).

Section 2-Reviewer 1 comments to the authors:

Author response: First of all, we would like to thank you for all the relevant comments and suggestions that were essential for the manuscript improvement.

a) Supplementary material table, is difficult to follow since it is based on the patient ID rather than in a field related to the topic of the MS (i.e. pathogenic variant, or IHC o whatever is chosen)

Author response: We rearranged the table´s information in alphabetic order according to the gene harboring the pathogenic variant described to facilitate comparison. The table was also included in the main text as Table 2.

b) In Figure 2. a straight category might be more adequate (like “Benign”) rather that “No PV or VUS”

Author response: “no PV or VUS” group was renamed as “no variants”.

c) There are representative studies in Brazil that were ignored in this MS, and the utility is obvious to contrasting the results of the cohorts. Although the authors disclosed:

This study had several limitations. It was retrospective and had a limited sample size. However, it represents the largest Brazilian BC cohort from a single institution tested using a multigene panel for hereditary BC (12,13). These findings require validation in other cohorts. The study was conducted at a private cancer center and, for this reason, the assessed population may differ from that of the general community. Furthermore, it consisted of a high-risk population for hereditary cancer with a median age of 45 years at BC diagnosis, mostly composed of patients with premenopausal BC (68.7%), a positive family history for cancer (89.7%), and a personal history of multiple primary cancers (17.4%).

It might be a good improvement to discuss comparing with larger cohorts, even of other regions, from the large country Brazil.

Author response: We agree with your comment. Nevertheless, most germline breast cancer studies in the Latin America, including Brazil, focused on BRCA genes and used different testing approaches (founder mutation, single gene analysis, copy number variation evaluation, multigene panel testing). We prepared a table, which was presented above (Section 1- Editor´s comments to the authors, comment number 4) to try to illustrate the most relevant Brazilian studies and multigene testing from Brazil and nearby countries. 

As illustrated in the table, the Brazilian studies used different approaches of genetic testing. For this reason, it was difficult to compare results. Besides this, most Brazilian studies were conducted in the south and southeast of the country. 

The best scenario would be the comparison of our data to other multigene panel testing studies. Pubmed search pointed out only 3 studies with multigene panel testing (2 from Brazil and 1 from Colombia). We rewrote the discussion trying to incorporate suggestions.

There was an intense racial interaction in Brazil, mainly Portuguese, African and Amerindian. There was also the influence of various flows of migrants coming to Brazil in the nineteenth and twentieth centuries, including French, Germans, Italians, Japanese, Middle Easterners (Turks and Arabs), Polish, Russians and Spanish. For this reason, each region has a different ancestry composition. 

d) Discussing the application of the NCCN guidelines upon the findings in the present work, may be interesting

Author response: The indication of genetic testing according to the NCCN criteria was added to the results and to the discussion.

Changes in the text of the revised manuscript (Results): page 10 line 142-144, “A total of 191 patients (85.3%, 191/224) fulfilled NCCN hereditary BC testing criteria. Among these patients, 23.5% harbored PVs (45/191). In the group of patients that did not meet NCCN criteria, PV detection rate was 3% (1/33) (Appendix 3 in the Supplement).”

Changes in the text of the revised manuscript (Discussion): page 12 line 209-217, “The detection rate of actionable PVs varies widely depending on the selected studied population and the genetic testing approach (founder mutations, single gene analysis, copy number variation evaluation, multigene panel testing). The present cohort consisted of consecutive BC patients referred to genetic counseling selected due to the suspicion of hereditary BC. The overall detection rate of PVs was 20.5% (46/224). According to NCCN criteria, 85.3% (191/224) of our cohort met hereditary BC testing criteria, among these patients 23.6% (45/191) harbored PVs. Whereas, among 33 patients that did not met NCCN criteria, only one harbored a PV (3%). The detection rate of PVs was similar to other multigene panel testing studies based on hereditary breast and ovarian cancer (HBOC) criteria (5,12,13,30–32).”

e) A good idea is to deposit the variants in a public database for the free access to the worldwide scientific community

Author response: All germline genetic tests were performed by commercial molecular diagnostic laboratories, this information is detailed the S1 (supplement material). The majority of the genetic tests (196/219, 90%) were performed in the laboratory Invitae (San Francisco, California, USA). The variants are included in publicly accessible databases such as SNPdb and ClinVar by the commercial laboratories policies. We also included in the Table 2 and S4 the rsID related to the Single Nucleotide Polymorphism Database identification or the Clinvar variant identifier number (variation ID) of each detected pathogenic/likely pathogenic variant or variant of uncertain significance.

Section 3-Reviewer 2 comments to the authors:

Author response: First of all, we would like to thank you for all the relevant comments and suggestions that were essential for the manuscript improvement.

Reviewer #2: Methods: I would recommend using primary breast cancer as an inclusion criterion.

Author response: The National Comprehensive Cancer Network guidelines for Genetic/Familial High-Risk Assessment (https://www.nccn.org/professionals/physician_gls/default.aspx) include both invasive and ductal carcinoma in situ breast cancers as a criteria for further genetic evaluation (footnote d, version 1.2021, page 17 from NCCN guideline). 

Results:

Study population: I would recommend excluding the remaining sixteen patients who had previously been diagnosed with other cancer because according to the title this paper is focused on breast cancer.

Author response: Multiple primary cancers is one of the most important clinical criteria for cancer predisposition syndromes screening. For this reason, the remaining sixteen patients that had been previously diagnosed with cancer, including thyroid cancer (6/16), melanoma (4/16), central nervous system (2/16), kidney (1/16), uterine cancer (1/16), lymphoma (1/16), and pancreatic cancer (1/16), were not excluded from the study population. Melanoma and pancreatic cancers are included in BRCA tumor spectrum. Uterine cancer (not specified as endometrial or cervix) as well as kidney cancer are included in Cowden syndrome tumor spectrum. Central nervous system, lymphoma and thyroid cancers are included in the Li-Fraumeni syndrome spectrum. Furthermore, four of these patients (25%, 4/16) harbored pathogenic variants. Multiple primary cancers was a criteria statistically significantly correlated with pathogenic variant detection.

Table 1: I would recommend including an extra column with the information of the total cohort for each item.

Author response: Total cohort is expressed in the edge of each column. We added the missing data information of each item. 

Frequency and Spectrum of Pathogenic Germline Variants: I recommend including in supplementary data analyzed genes in each group of patients. I would also recommend presenting in the article text a table with the PV detected.

Author response: We added in the article a table with the PVs detected (Table 2), as recommended. We added as Appendix 1 all the information about the performed genetic tests: commercial laboratories, number of genes in each panel, minimum of genes included in all the panels. The majority of the genetic tests (196/219, 90%) were performed in the commercial laboratory Invitae (San Francisco, California, USA), and included 80-84 genes.

Correlation between test positivity, germline genotype, and clinical data: I recommend including a table with the VUS detected in supplementary data.

Author response: We added a table with VUS detected in the supplementary material (S4) as recommended.

---

## [Decision Letter · Decision Letter 1]

25 Jan 2021

PONE-D-20-32343R1

Germline molecular data in hereditary breast cancer in Brazil: lessons from a large single-center analysis.

PLOS ONE

Dear Dr. Sandoval,

Thank you for submitting your manuscript to PLOS ONE. After careful consideration, we feel that it has merit but does not fully meet PLOS ONE’s publication criteria as it currently stands. Therefore, we invite you to submit a revised version of the manuscript that addresses the points raised during the review process.

1. Address the two minor points raised by reviewer 1.

We look forward to receiving your revised manuscript.

Kind regards,

Amanda Ewart Toland, Ph.D.

Academic Editor

PLOS ONE

Reviewers' comments:

Reviewer's Responses to Questions

**Comments to the Author**

1. If the authors have adequately addressed your comments raised in a previous round of review and you feel that this manuscript is now acceptable for publication, you may indicate that here to bypass the “Comments to the Author” section, enter your conflict of interest statement in the “Confidential to Editor” section, and submit your "Accept" recommendation.

Reviewer #1: All comments have been addressed

Reviewer #2: All comments have been addressed

2. Is the manuscript technically sound, and do the data support the conclusions?

Reviewer #1: Yes

Reviewer #2: Partly

3. Has the statistical analysis been performed appropriately and rigorously? 

Reviewer #1: Yes

Reviewer #2: I Don't Know

4. Have the authors made all data underlying the findings in their manuscript fully available?

Reviewer #1: Yes

Reviewer #2: No

5. Is the manuscript presented in an intelligible fashion and written in standard English?

Reviewer #1: Yes

Reviewer #2: Yes

6. Review Comments to the Author

Reviewer #1: The MS “Germline molecular data in hereditary breast cancer in Brazil: lessons from a large single-center analysis” by Renata Lazari Sandoval, M.D., M.Sc et al, have made an amount of corrections.

1. One point that needs to be repaired is the sentence in page 14: “Even though most of the published data on hereditary BC among Latin American countries are from Brazil [11]” referring to the reference 11, should be rephrased since it might induce to a wrong concept that much of the contribution in Latin America is by the number of publications from Brazil, which is wrong. That review it is associated to PARP inhibitors and immunotherapy (stated in the title) which, besides, it is not reflecting the sense of this sentence. More important in that review, among most of the papers from Brazil, few ones refer to genes’ panels (a concept immediately expressed in the MS), many of the referenced Brazilian papers are biased by methodology limitations or patients´ selection, many refers to TP53 solely and, to be rescued, two references reflect the meaning of the sentence (ref 42 and ref 60). Importantly, published works in Latin America referred in the same review reflects the experience in a large series of cases that no one has reached at the year of publication (ref 98 and 100), these four publications are the ones that deserve a comment if that sentence is kept.

2. A reference should be added in the sentence pg. 26 line 263: “Our results are in concordance with previous reports stating that 13% of patients with multiple primary BC harbor PVs in BRCA2, CHEK2, BARD1, and RAD51C

Reviewer #2: (No Response)

7. PLOS authors have the option to publish the peer review history of their article (what does this mean?). If published, this will include your full peer review and any attached files.

Reviewer #1: **Yes: **Angela SOLANO

Reviewer #2: **Yes: **Laura Cifuentes-C

---

## [Author Response · Author response to Decision Letter 1]

4 Feb 2021

Dear Dr. Toland, 

Thank you for giving us the opportunity to submit a second revised version of the manuscript “Germline molecular data in hereditary cancer in Brazil: lessons from a large single-center analysis”. We are grateful for the time and effort that you and the reviewers dedicated to providing feedback on our manuscript. All the comments provided the possibility of relevant improvements to our manuscript.

We have incorporated the suggestions made by the reviewer1. Authors´ responses are described in blue. Changes performed in the revised manuscript are highlighted in yellow. Page numbers refer to the revised manuscript file with tracked changes.

Reviewer #1: 

1. One point that needs to be repaired is the sentence in page 3: “Even though most of the published data on hereditary BC among Latin American countries are from Brazil [11]” referring to the reference 11, should be rephrased since it might induce to a wrong concept that much of the contribution in Latin America is by the number of publications from Brazil, which is wrong. That review it is associated to PARP inhibitors and immunotherapy (stated in the title) which, besides, it is not reflecting the sense of this sentence. More important in that review, among most of the papers from Brazil, few ones refer to genes’ panels (a concept immediately expressed in the MS), many of the referenced Brazilian papers are biased by methodology limitations or patients´ selection, many refers to TP53 solely and, to be rescued, two references reflect the meaning of the sentence (ref 42 and ref 60). Importantly, published works in Latin America referred in the same review reflects the experience in a large series of cases that no one has reached at the year of publication (ref 98 and 100), these four publications are the ones that deserve a comment if that sentence is kept.

Author response: We fully agree and apologize for this misunderstanding. This comment was very important. We rephrased the sentence and included two references.

Changes in the text of the revised manuscript: page 3 line 40-41, “Even though there are published data on hereditary BC among Latin American countries [11,12], few studies have included non-BRCA genes and multigene panel testing [13,14]. Brazil is the largest country in South America and is the most genetically heterogeneous [14]. Most studies performed in Brazilian cohorts on hereditary breast cancer have been performed in the Southern and Southeastern parts of the country [12,15]. For this reason, epidemiological data about the prevalence of cancer predisposition syndromes in other parts of the country are needed.”

2. A reference should be added in the sentence pg. 26 line 263: “Our results are in concordance with previous reports stating that 13% of patients with multiple primary BC harbor PVs in BRCA2, CHEK2, BARD1, and RAD51C

Author response: Thank you very much for this comment. We added this missing information and rephrased the data to better express our results in comparison to the literature. In our cohort 13/224 (5.8%) of patients had bilateral breast cancer (BC), seven patients did not harbor pathogenic variants (PVs). Six patients harbored PVs (6/13, 46%). Four out of thirteen (31%) patients harbored a PV in non-BRCA genes. Shin et al. (2020) described a cohort (n=496) with 15% of bilateral BC, 31,6% harbored PVs in non-BRCA genes (CHEK2, TP53, NBN, CDH1, MRE11A). Fanale et al. (2020) suggested that all bilateral BC patients should be offered multigene testing. They analysed a cohort of 139 cases of bilateral BC, 35.8% (19/53) harbored PVs in non-BRCA genes (PTEN, PALB2, CHEK2, ATM, RAD51C).

Changes in the text of the revised manuscript: page 15 line 262-266, “Patients with cancer predisposition syndromes have a higher risk of developing a second primary BC, especially for BRCA 1/2 mutation carriers [45]. Nevertheless, 8 - 36% of patients with bilateral BC harbor PVs in other genes beyond BRCA [32,47,48]. Bilateral BC represented 5.8% (13/224) of our cohort. Thirty-one percent (4/13) of patients with bilateral BC harbored PVs in non-BRCA genes (CHEK2, BARD1 and RAD51C). These results are in concordance with previous studies [32,47].”

---

## [Editor Report · Decision Letter 2]

8 Feb 2021

Germline molecular data in hereditary breast cancer in Brazil: lessons from a large single-center analysis.

PONE-D-20-32343R2

Dear Dr. Sandoval,

We’re pleased to inform you that your manuscript has been judged scientifically suitable for publication and will be formally accepted for publication once it meets all outstanding technical requirements.

Kind regards,

Amanda Ewart Toland, Ph.D.

Academic Editor

PLOS ONE
---

## [Editor Report · Acceptance letter]

10 Feb 2021

PONE-D-20-32343R2 

Germline molecular data in hereditary breast cancer in Brazil: lessons from a large single-center analysis. 

Dear Dr. Sandoval:

I'm pleased to inform you that your manuscript has been deemed suitable for publication in PLOS ONE. Congratulations! Your manuscript is now with our production department. 

Kind regards, 

on behalf of

Dr. Amanda Ewart Toland 

Academic Editor

PLOS ONE